# Modeling of Fatigue Wear of Viscoelastic Coatings

**DOI:** 10.3390/ma14216513

**Published:** 2021-10-29

**Authors:** Fedor I. Stepanov, Elena V. Torskaya

**Affiliations:** Ishlinsky Institute for Problems in Mechanics, Russian Academy of Sciences, 119526 Moscow, Russia; stepanov_ipm@mail.ru

**Keywords:** contact problem, sliding, viscoelasticity, coating, damage, fatigue wear

## Abstract

A new model for studying the kinetics of fatigue wear of a viscoelastic coating bonded to a rigid substrate is proposed. The fatigue mechanism is due to the cyclic interaction of the coating with a rough counterbody, which is modeled by a periodic system of smooth indenters. The study includes the solution of the problem of sliding contact of the indenter at a constant velocity along the viscoelastic coating, the calculation of stresses taking into account the mutual effect, and study of the process of damage accumulation in the material. The calculation of the damage function of the surface layer was carried out using the reduced stress criterion. Assuming the possibility of summation of accumulated damage, two processes were considered: delamination of surface layers of the coating and continuous fracture of the surface by the fatigue mechanism. The effect of the sliding velocity and viscoelastic properties of the material on the damage accumulation and the coating wear rate was analyzed. Two types of load, constant and stochastically varying, were used in modeling and analysis. It was found that the rate of fatigue wear of the coating increased and then became constant.

## 1. Introduction

Modeling and experimental study of the wear of materials with rheological properties is important for predicting the durability of friction units operating under specified conditions. These materials (rubbers, for example) are often used as coatings to decrease contact stiffness, reduce noise, etc.

The first publications in which it was established that one of the main mechanisms of polymer wear is contact fatigue appeared in 1963–1966 [1,2,3,4,5,6]. The fatigue nature of wear was also confirmed by other authors [7,8]. Fatigue wear is the least intense type of wear, but for a number of friction units it is the main one.

For the fatigue type of fracture, it is very important to know the criterion of damage accumulation. The results of classical fatigue tests, especially those obtained under uniaxial loading, can potentially be used to simulate fatigue wear. In the experimental study of fatigue phenomena under uniaxial loading and torsion, deformations are most often chosen as criteria for the initiation of fatigue defects, since they can be easily determined experimentally [9] (by measuring displacements). Maximal principal strain [10,11,12] and octahedral tangential strain [9] are also widely used. In early studies [10] on fatigue of rubber, it was estimated that increasing of the minimal strain up to some certain value could increase the fatigue life. Later, it was found [13,14] that under the same values of maximum principal strain, the fatigue life in the case of simple tension differs from that in the case of equibiaxial tension. Thus, the principal strain criterion is not appropriate for fatigue life prediction in case of multiaxial loading [15]. Stress-based criteria, while more difficult to obtain in experiments than strain-based criteria, appear to be more efficient in prediction of fatigue life under multiaxial loading [16,17,18]. It was found that the plane of a crack is perpendicular to the maximal principal stress.

Another criterion for fatigue prediction is the strain energy density, which is based on an approach that considers the process of crack evolution. It was found that under certain conditions, energy release rate is proportional to strain energy density [19,20]. While more efficient than maximal principal strain at fatigue life prediction [15,17], strain energy density still cannot adequately describe the relation between uniaxial tension and multiaxial loading [13,14]. The cracking energy density proposed by Mars [21,22] for predicting crack nucleation and orientation appears to be a more adequate parameter than strain energy density [18]. In [23], the authors proposed an efficient model for prediction of fatigue life based on continuum damage mechanics [24] combined with cracking energy density. The model was verified by tension-torsion experiments on sulfur-vulcanized SBR and shown to be efficient. The capability of different fatigue criteria to unify experimental results on a carbon-filled styrene–butadiene rubber (SBR) material was checked in [25]. Cracking energy density- based and stress-based criteria were found to be the most effective. A more comprehensive review on papers using conventional approaches to fatigue modelling of rubbers can be found in [26]. The deep learning algorithms that are currently widely used in many engineering problems found an application in modeling of fatigue life prediction [27,28,29]. The development of these approaches is intended to overcome the limitations of conventional models in regard to specific materials and loading conditions. Nevertheless, further research is necessary to prove reliability and rationality of these methods [27].

Experiments with the sliding contact of a sphere with a rubber disc were described and analyzed in [5]. Since this paper about this fundamental research was published in Russian a very long time ago, its main points (materials, methods, and results) are summarized in Appendix A. It was found that at first, a small friction track appeared on the disc; then, the indenter slid for a long time without significant changes; and finally, after a certain number of cycles, the most intensive separation of wear particles began. Thus, there was a phase of damage accumulation (the so-called incubation period) and a phase of intense wear. A correlation was found between the experimental results on frictional contact fatigue and standard fatigue tests under cyclic tension (the curves of the dependence of the number of cycles up to the moment of failure on the applied load were parallel; see Appendix A).

Fatigue wear was studied in experiments with sliding contact of metal balls and various types of polymers [30] (polycarbonate, polyvinyl chloride, ultra-high molecular weight polyethylene, etc.). The results showed that the number of cycles before the start of wear was inversely proportional to the ratio of the tensile stress in contact to the tensile yield strength. The wear rate after the incubation period was found to be proportional to the modulus of elasticity.

In the theory of resistance of materials, among other criteria, stress reduced to simple tension was considered [31]. It was included in the fatigue fracture criterion in the experiments of Kragelsky and Nepomnyashchy [5]; they used the contact problem solution obtained by Saverin [32].

The main reason for the fatigue fracture of the surface layers is the roughness of the surfaces of the contacting bodies, which leads to a cyclically changing stress field in the surface layers of the material. In the presence of other types of wear, such as abrasive or adhesive wear, fatigue wear can affect the overall wear rate. There is also an option when fatigue damage does not have time to accumulate, i.e., the damaged material wears out until the damage reaches a critical value [33]. In the case of elastomers, fatigue wear can be the main failure mechanism (for example, through rolling contact).

Approaches to modeling the fatigue wear of elastic materials were described in [34]. Wear models based on the criterion of maximum shear stresses were described in [35] for an elastic half-space and in [36] for materials with elastic coatings. Fatigue wear of viscoelastic half-space was considered in [37] using the criterion of reduced stresses described above. Removal of material without changing the shape of the surface of the half-space occurs while maintaining contact and internal stresses. Fatigue wear of coatings (decreases in their thickness) is accompanied a change in internal stresses, since the total rigidity of the system increases.

This paper presents the results of modeling the fatigue wear of relatively compliant coatings made of viscoelastic materials, based on the criterion of reduced stresses. This study is a logical continuation of modeling the fatigue wear of elastic layered bodies [36] and viscoelastic half-space [37].

## 2. Steps of Fatigue Wear Modeling

Under cyclic loading of the surface, which occurs with relative displacements of rough bodies, an inhomogeneous cyclic field of internal stresses with high amplitude values arises in the surface layer, which is the reason for the accumulation of damage in this layer. A macroscopic approach for studying the damage accumulation process was originally developed in relation to standard fatigue tests and is described, for example, in [38]. It was found by Goryacheva [34] that this approach can be used for modeling fatigue wear. To simulate the fracture of a viscoelastic coating due to cyclic loading, the following algorithm [34,36] is proposed:calculation of contact pressure for a two-layer half-space with viscoelastic upper layer;calculation of internal stresses, taking into account the forces of friction for multiple contact;selection of a function that connects the rate of damage accumulation with stresses inside the viscoelastic layer;calculation of damage;analysis of fracture kinetics.

Internal stresses depend on the following parameters [39]: the viscoelastic properties of the layer material, the layer thickness, parameters characterizing the geometry of the counterbody, and the load. If the stochastic nature of loading [34] is considered, the range of load variation is also a necessary parameter.

Calculation of damage and fracture kinetics involves the study of processes that depend on time, the strength characteristics of the coating material, and the time-changing stress state.

A simplified contact model was used in this study. This model, on the one hand, took into account the main features of the contact interaction (leading to the formation of fatigue damage in the subsurface layers of the material and to wear). On the other hand, it allowed us to study the effects of the main parameters of the process (the mechanical and strength properties of the interacting bodies, the microgeometry of their surfaces, load and sliding velocity) on the wear rate of the coating by the fatigue mechanism.

## 3. Problem Formulation (Contact Problem and Internal Stresses)

A periodic system of spherical indenters with radius *R* located at the nodes of a hexagonal lattice with period *l* was considered as a model of a rough surface. The system of indenters moved at a constant velocity along the boundary of the viscoelastic coating (Figure 1). The direction of sliding was the same as the direction of the axis Ox. The pressure averaged over the period was $p_{n}$. Shear stresses balanced the frictional forces acting in the contact zones where the Amontons–Coulomb law was valid ($\tau_{n}=\mu p_{n}$, where μ is the coefficient of sliding friction).

When solving the contact problem, it was assumed that the effect of surface shear stresses caused by friction on normal stresses and displacements could be neglected. In [40], the influence of shear stresses on the solution of a contact problem for a slider and a viscoelastic half-space was analyzed. It was shown that this effect was negligible for weakly compressible materials.

In this study, the contact problem was solved for a single slider without taking into account the mutual effect (influence of neighboring indenters on contact pressures under a selected indenter). The effect in the contact of sliders with a viscoelastic half-space was studied in [36,41]. In the case of a viscoelastic layer, the lower the layer thickness, the lesser the mutual effect is [42].

The sliding of a rigid smooth body over the surface of a viscoelastic layer of thickness *h* was under consideration; the layer was bonded to the rigid half-space.

The smooth slider, loaded with vertical force $P=\sqrt{3}p_{n}l^{2}/2$, moved at a constant velocity in the direction Ox. The moving coordinate system was associated with the slider. The problem was quasistatic; stresses and displacements were time independent.

We considered the following boundary conditions at the surface:(1)$$\begin{array}{l} w(x,y,0)=f(x,y)+D,\;\;(x,y)\in \Omega ;\\ \sigma_{z}(x,y,0)=0,\;\;(x,y)\notin \Omega ;\\ \tau_{xz}(x,y,0)=\tau_{xy}(x,y,0)=0,\;\;-\infty <(x,y)<+\infty . \end{array}$$

Here, Ω is the contact zone; w(x,y,0) are surface normal displacements, D is the slider penetration; and $\sigma_{z}$, $\tau_{xz}$, and $\tau_{yz}$ are normal and shear stresses. The slider shape was defined by a smooth function f(x,y). Contact zone Ω and pressure distribution $p(x,y)=-\sigma_{z}(x,y,0)$ are initially unknown.

We used also the equilibrium condition
(2)$$P=\iint_{\Omega }p(x,y)dxdy$$
and the condition that the pressure on the boundary of the contact zone was equal to zero, corresponding to a smooth indenter.

The conditions at the interface (z=h) corresponded to complete adhesion:(3)$$w(x,y,h)=0,\;u_{x}(x,y,h)=0,\;u_{y}(x,y,h)=0$$

Here, $u_{x}(x,y,z)$ and $u_{y}(x,y,z)$ are tangential displacements of the layer.

The material model was described by an integral operator [40] connecting shear strains γ(t) with stresses τ(t):(4)$$\gamma (t)=\frac{1}{G}\tau_{xz}(t)+\frac{1}{G}\int_{-\infty }^{t}\tau (t)K(t-\tau )d\tau$$

Here, G is the instantaneous shear modulus; the creep kernel is the following:(5)$$K(t)=k\exp\left(-\frac{t}{\omega }\right)$$

Here, ω is the retardation time, and k is the reciprocal of the relaxation time. Poisson’s ratio ν was assumed to be constant.

The method of the contact problem solution was based on double integral Fourier transforms and an iterative procedure [43].

The resulting pressure distribution was used to calculate internal stresses in the viscoelastic layer. Shear stresses in the contact area were added to the boundary conditions:(6)$$\tau_{xz}(x,y)=\mu p(x,y),\;\;(x,y)\in \Omega$$

Stresses in a layered elastic half-space depend only on Poisson’s ratio [44], which is constant for the considered viscoelastic layer model. Therefore, the formulas for the elastic layer could be used for the viscoelastic one. Since the system of indenters was considered, the stress state in the coating was determined while accounting for the mutual influence of indenters by the superposition method [36,45]. By summing over all loaded surface features, we could determine the stress components at any point within the layer.

## 4. Model of Damage Accumulation and Fatigue Wear of Coating

To study the accumulation of damage, characterized by a function Q(x,y,z,t) that does not decrease in time, the model of linear summation of damage was used (at each moment of time, the increase in damage does not depend on the history of the process) [34]. Fracture was considered to start at time t* when Q(x,y,z,t) obtains a predetermined critical value.

The damage accumulation rate depends on the contact conditions as well as on the material properties. The choice of a physical approach to modeling damage is usually based on experimental data. In the case of elastomers, the results obtained in [5] were classical and could be used to develop a damage accumulation model for the material under the surface. In this study, the number of cycles before the initiation of the fatigue crack was connected with the maximal values of reduced stresses. According to the hypothesis of damage linear summation, we had the following relation for the rate of damage accumulation [19]:(7)$$q(x,y,z,t)=\frac{\partial Q(x,y,z,t)}{\partial t}=g{\left(\sigma_{p}(x,y,z,t)\right)}^{m}$$

Here, g and m are experimentally determined constants, and $\sigma_{p}(x,y,z,t)$ are the reduced stresses. To calculate the reduced stresses [37], after determining all stress components [36,46], we found the main stresses at the point:(8)$$\det\left|\begin{array}{ccc} \sigma_{x}-\sigma & \tau_{xy} & \tau_{xz}\\ \tau_{yx} & \sigma_{y}-\sigma & \tau_{yz}\\ \tau_{zx} & \tau_{zy} & \sigma_{z}-\sigma \end{array}\right|=0$$

Then, using the roots of the equation ($\sigma_{1}>\sigma_{2}>\sigma_{3}$), the reduced stress was determined:(9)$$\sigma_{p}=\frac{1}{\sqrt{2}}\sqrt{{\left(\sigma_{1}-\sigma_{2}\right)}^{2}+{\left(\sigma_{2}-\sigma_{3}\right)}^{2}+{\left(\sigma_{3}-\sigma_{1}\right)}^{2}}$$

By virtue of the problem’s periodicity, the damage function did not depend on x and y coordinates, being a function of z coordinate and time t. Time in the damage function can be replaced by a number of cycles N as Q=Q(z,N).

It is possible to obtain damage Q accumulated in an arbitrary point z after N cycles using the following relationship, which is derived from (7):(10)$$Q(z,N)=\int_{0}^{N}q_{n}(z,n)\Delta tdn+Q_{0}(x,y,z)$$

Here, $Q_{0}(x,y,z)$ is the distribution of the initial damage to the material, Δt is the time of one cycle, and $q_{n}(z,n)$ is the rate of damage accumulation not depending on coordinates x,y.

The fracture was considered to occur when the damage in a point reached the critical value. This condition can be written in a normalized system as the following:(11)Q(z,N*)=1
where N* is the number of cycles before the fracture.

Calculation of stress distribution within the layer allowed obtaining the maximal values of reduced stresses along the Ox axis, which coincides with sliding direction of the system of indenters. Maximal values of ${\tilde{\sigma }}_{p}(z)$ occurred in a plane, which is a cross-section of the layer including the center of a contact spot.

The number of cycles before a fracture can be obtained with the following expression based on (7), (10), and (11):(12)$$\int_{0}^{N*}g{\left({\tilde{\sigma }}_{p}(z,n)\right)}^{m}\Delta tdn+Q_{0}(x,y,z)=1$$

If there is no initial damage, it is possible to calculate a number of cycles before crack initiation at or under the surface, where the reduced stress reaches its maximum value. For this one, can use the following expression derived from (12):(13)$$\begin{array}{l} N*={\left(g\Delta t{\left(\max{\tilde{\sigma }}_{p}(z,n)\right)}^{m}\right)}^{-1}\\ Q*(z)=N*g\Delta t{\left({\tilde{\sigma }}_{p}(z,n)\right)}^{m},\;z\leq -h \end{array}$$

Here, Q* is the damage that needs to be considered in further study of the damage accumulation.

If maximal damage is located under the surface, the function Q*(z) obtains the critical value on the newly formed surface after the material delamination and removal. Surface fatigue was considered to occur after the first act of fracturing. The function ${\tilde{\sigma }}_{p}(z,0)$ may reach its maximum on the surface. In this case, the acts of material removal correspond to the step of the grid used in calculations.

Analyzing the results obtained in [5], one can conclude that expression (7) describes well the fatigue phenomena for the three rubbers. The parameter of those *m* materials was almost the same and close to 0.3, while the parameter g significantly differed.

In reality, rough surfaces rarely have structures that can be modeled by a single-level system of indenters. Factors such as wide range of asperity heights, waviness of surfaces, and different types of vibrations taking place at frictional contact do affect the process of damage accumulation. In order to study these effects, one should consider loading with a variable value of nominal pressure $p_{n}(t)$. Results obtained for the case of homogeneous elastic half-space and probabilistic distributions of loading explained some data in wear experiments [35]. Therefore, to study the process of fatigue damage accumulation, a probabilistic load distribution was considered. An example of nominal pressure distribution in time is shown in Figure 2. The load change was provided by using a random number generator.

To carry out the study, it was necessary to discretize the continuous process of damage accumulation for each chosen $p_{n}(t)$ value. The following matrix was obtained after calculating the functions ${\tilde{\sigma }}_{p}(z,H-z\prime (t),p_{n}(t))$:(14)$$\left(\begin{array}{ccccc} {\left({\tilde{\sigma }}_{p}\right)}_{\max}^{\left(1,1\right)} & 0 & & & \\ {\left({\tilde{\sigma }}_{p}\right)}_{\max}^{\left(2,1\right)} & {\left({\tilde{\sigma }}_{p}\right)}_{\max}^{\left(2,2\right)} & 0 & 0 & \\ . & . & . & & \\ . & . & . & . & \\ {\left({\tilde{\sigma }}_{p}\right)}_{\max}^{\left(NN,1\right)} & . & . & . & {\left({\tilde{\sigma }}_{p}\right)}_{\max}^{\left(NN,NN\right)} \end{array}\right)$$

Here, NN is a number of points within the range 0–*H* (*H* is the initial thickness of the layer), which defines the points for calculating the differential of reduced stresses and the damage function. The first column of the matrix corresponds to the differential of reduced stresses within the layer with thickness *H*. The last column of the matrix, containing only one nonzero element, corresponds to the case of a layer worn out to the minimal thickness H/NN. The number of matrixes is defined by the step of discretized variation of pressure and by the chosen amplitude of $p_{n}$.

Function $z^{\prime }(t)$ defines surface displacement occurring due to fatigue:(15)$$z^{\prime }(t)=\Delta H(Q)$$

Specially written programs were used for calculations. The results were visualized using Maple.

## 5. Results of Calculations with Time-Independent Load

The following set of dimensionless parameters was used for calculation and analysis: coordinates $(x^{\prime },y^{\prime })=\left(x,y\right)/R$, velocity $V^{\prime }=V\omega /R$, layer thickness $h^{\prime }=H/R$, nominal pressure $p_{n}{}^{\prime }=p_{n}/E$, and contact pressure $p^{\prime }(x,y)=p(x,y)/E$. Viscoelastic properties of the material were characterized by the parameter c=kω, which is the relation of retardation time to relaxation time. This parameter for viscoelastic materials can vary over a wide range [47,48]; its value for the same material depends, for example, on temperature. The friction coefficient μ was also one of the parameters.

Below we present the results of our analysis of the effect of sliding velocity on contact pressure, the distribution of reduced stresses in the coating, the function of damage at different stages of the wear process, and the kinetics of wear (a decrease in the layer thickness). Calculations were carried out for the following values of the parameters: ν=0.4, ${p_{n}}^{\prime }=4.2\cdot 10^{-2}$, $h^{\prime }=0.667$ (initial layer thickness). The value of Poisson ratio is typical for polymers. The initial layer thickness was relatively large (the initial contact pressures were close to the appropriate solutions for the viscoelastic half-space with the same properties), which allowed us to analyze more steps of the wear process.

The selected velocities provided a significant difference in the distribution of contact pressure (Figure 3); the maximum pressure value was almost one and a half times higher for the case of the higher velocity.

To calculate the wear kinetics, it is necessary to recalculate the contact characteristics and internal stresses at each step of the calculations with a decrease in the coating thickness. Figure 4 illustrates how significantly the stresses changed with decreasing coating thickness. It presents the distributions of reduced stresses related to the maximal value of the contact pressure $p_{0}$ for the initial coating thickness (a) and with a threefold decrease in thickness (b). The results were obtained in the absence of friction.

In this particular case, there were two local maxima of reduced stresses (surface and subsurface). In the case of a thicker coating, the surface maximum is greater; in the case of a thinner coating, the subsurface maximum is greater. The minimum of the stresses was in any case on the surface, which was due to the mutual influence of the stress fields under each indenter. The maximum values of contact pressures in the case of a thinner layer were 1.2 times higher than in the case of a thicker layer.

The presence of friction forces significantly changed the stress distribution. In all the cases considered, at μ=0.3, the maximum values of the reduced stresses were localized on the surface. Figure 5 shows the dependences of the maximum value of reduced stresses on the surface $\sigma_{p}^{\left(m\right)}$ on the layer thickness in the absence of friction (curve 1) and at μ=0.3 (curve 2). The stresses were here related to the nominal pressure. It should be noted that (in comparison with a single indenter) the mutual effect somewhat reduced the value of the stresses under consideration, especially for thin layers.

To analyze the kinetics of fatigue wear, the process of accumulation of contact-fatigue damage was considered step-by-step. The linear dependence of the damage on the parameter *g* in (12) allowed it to be included in a dimensionless complex $S^{\prime }=gN\Delta tl^{\prime }$ characterizing the friction path. The value of *m*, as noted in the previous section, can for some rubbers be taken as equal to 0.3.

Figure 6 shows the distributions of the damage function at different points in time. In the later stages of wear, the function becomes close to linear.

To analyze the effect of the sliding velocity, friction coefficient, and rheological properties of the layer material on the wear rate, the curves shown in Figure 7 were obtained. For convenience of comparison, the wear rate was related to the number of cycles in the incubation period (N*) before fracture, since the time of passage of one period was very different during sliding at different velocities. An increase in sliding velocity (Figure 7a) led to a decrease in the number of cycles in the incubation period and to faster wear (N* for curve 1 was less than that for curve 2). In this case, the further wear rate, related to N*, was higher for a lower velocity. The wear rate became almost constant shortly after the incubation period. Apparently, this was due to the fact that, along with an increase in the maximum values of the contact pressure during wear, there was an equalization of the stress values over the layer thickness. Predictably, wear began and occurred faster in the presence of friction forces (Figure 7b). N* in the presence of friction was 24% less. The rheological properties of the layer material were characterized by the parameter c. The higher the value of this parameter, the greater the rigidity of the layer was when the indenters slid over it, and the lesser the contact area and mutual influence were. The smaller the value of c, the greater the number of cycles before the start of fracture was. Comparison of Figure 7a,c shows that the effect of parameter c was qualitatively the same as the effect of velocity.

## 6. Results of Calculations with Stochastic Load

The distribution of damage over the thickness of the elastic layer at different times is shown in Figure 8 for cases of stochastic pressure changes with an amplitude of ±20% (Figure 8a) and ±80% (Figure 8b) of the average of the nominal pressure. These Figures can be compared with Figure 6, since the other input parameters of the calculation were the same. In the case of Figure 8a, the maximum damage was localized on the surface (as in Figure 6), but the number of cycles in the incubation period was slightly higher (by 8%). Also, damage was more evenly distributed throughout the layer thickness, which accelerated wear after the incubation period. With a more stochastic distribution of the nominal pressure over time, the process had another scenario. The maximum damage accumulated under the surface at a distance of about 0.08 *h* from the surface. This was due to the fact that the main contribution to the process was made by the impact of relatively large loads. In this case, the size of the contact area grew with respect to the layer thickness, that is, a stressed state close to that shown in Figure 4b was realized. After the incubation period, a layer of final thickness was delaminated, and then wear occurred from the surface. The more stochastic the loading, the longer the incubation period was.

The kinetics of the decrease in the layer thickness at different values of the nominal pressure amplitude can be analyzed according to the curves in Figure 9. Curve 1 coincided with curve 2 in Figure 6, and curves 2 and 3 were obtained for the same parameters as in Figure 8a,b, respectively. Here, the act of subsurface fracture with delamination of a part of coating is clearly demonstrated (curve 3).

It should be noted that a special area (in terms of fatigue resistance) is the coating–substrate interface. Damage [49] can also accumulate on the interface, which can cause the coating to delaminate before it wears out. In the case of a viscoelastic layer on a rigid substrate, the maximum shear stresses [39] are concentrated at the interface, which (under cyclic loads) can lead to disruption of the adhesion of the coating to the substrate.

## 7. Conclusions

A method for calculating contact-fatigue damage in a viscoelastic coating under sliding contact conditions was developed. The semianalytical method provided fast solutions to contact problems, which is very important for continuously changing coating thickness due to wear. The dependence of the wear rate on the input parameters of the problem was analyzed. The effects of the sliding velocity, the friction coefficient, and the rheological properties of the layer material on the time of fatigue damage initiation and wear rate were considered.

It was found that the higher the sliding velocity, the fewer cycles were needed to start fatigue fracture. The effect of the parameter *c*, which characterizes the rheological properties of the layer material, was the same. The presence of friction (nonzero friction coefficient) increased internal stresses and therefore the rate of fatigue wear.

It was found that the rate of wear from the surface after the incubation period gradually increased and then tended to be constant. The stochastic nature of loading (typical for real contact conditions) could lead to the delamination of a layer of finite thickness after the end of the incubation period. After that, wear occurred from the surface. The incubation period for stochastic loads was higher. Also, damage was more evenly distributed throughout the layer thickness, which accelerated wear after the incubation period.

## Figures and Tables

**Figure 1 materials-14-06513-f001:**
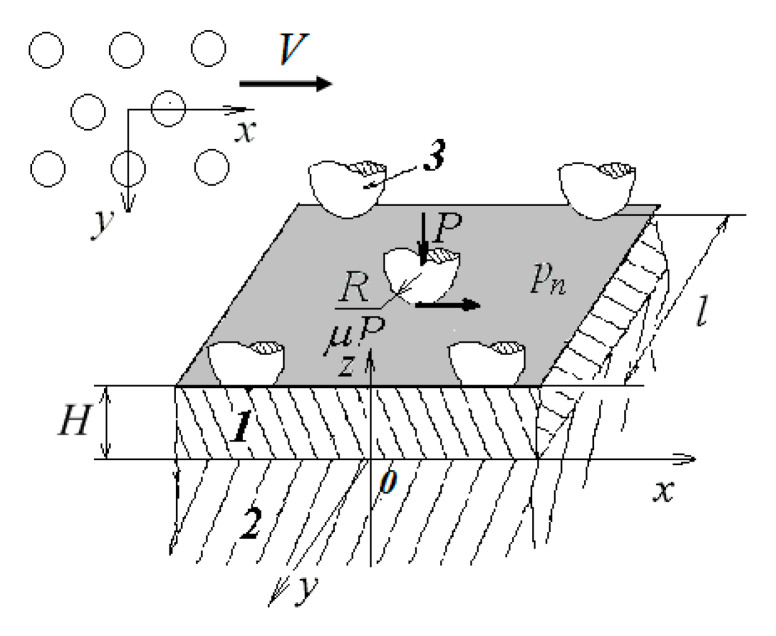
Scheme of multiple contact of viscoelastic coating (1) bonded to a rigid substrate (2) and a periodic system of indenters (3).

**Figure 2 materials-14-06513-f002:**
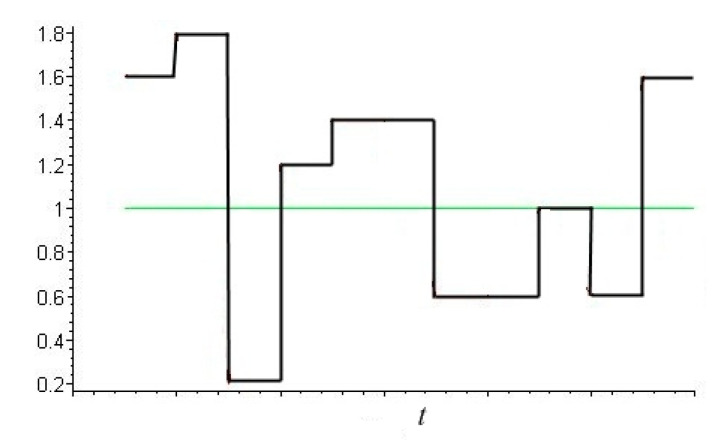
Example of stochastic variation of nominal pressure under frictional contact.

**Figure 3 materials-14-06513-f003:**
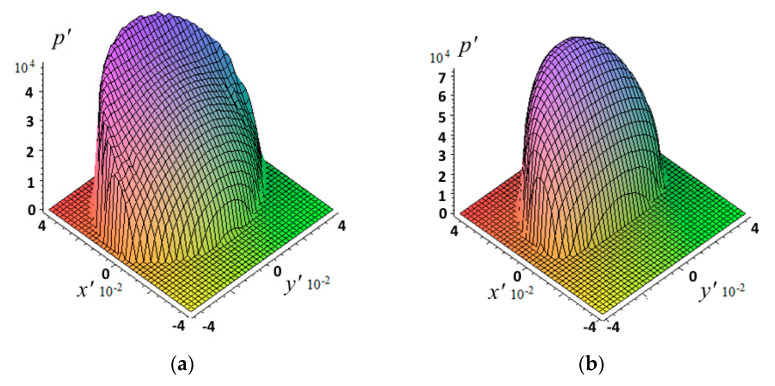
Contact pressure distribution during the sliding process: c=5, ν=0.4, ${p_{n}}^{\prime }=4.2\cdot 10^{-2}$, $h^{\prime }=0.667$, $V^{\prime }=0.0208$ (**a**) or $V^{\prime }=0.0625$ (**b**).

**Figure 4 materials-14-06513-f004:**
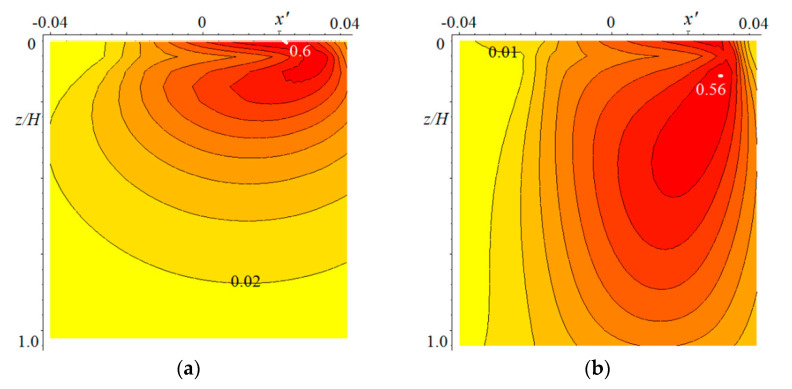
Distribution of the reduced stress: c=5, ν=0.4, ${p_{n}}^{\prime }=4.2\cdot 10^{-2}$, $V^{\prime }=0.0208$, $h^{\prime }=0.1$ (**a**) or $h^{\prime }=0.33$ (**b**).

**Figure 5 materials-14-06513-f005:**
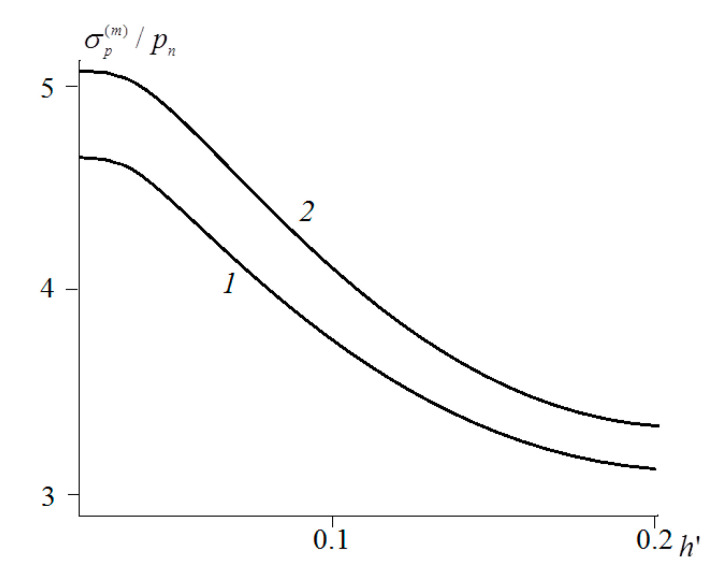
Dependence of the reduced-stresses maximum on the layer thickness: c=5, ν=0.4, ${p_{n}}^{\prime }=4.2\cdot 10^{-2}$, $V^{\prime }=0.0208$, μ=0 (curve 1) or μ=0.3 (curve 2).

**Figure 6 materials-14-06513-f006:**
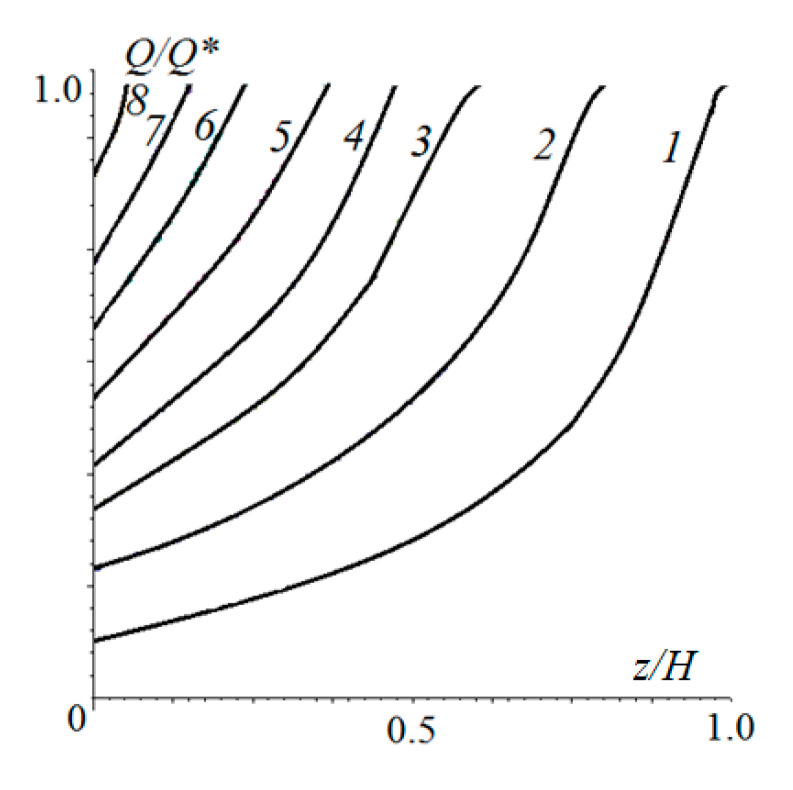
Distribution of the damage function within the layer at different phases of fatigue: μ=0, c=5, ν=0.4, ${p_{n}}^{\prime }=4.2\cdot 10^{-2}$, $h^{\prime }=0.667$, $V^{\prime }=0.0208$, N/N*=1, 1.35, 1.53, 1.63, 1.73, 1.80, 1.85, or 1.90 (curves 1–8, respectively).

**Figure 7 materials-14-06513-f007:**
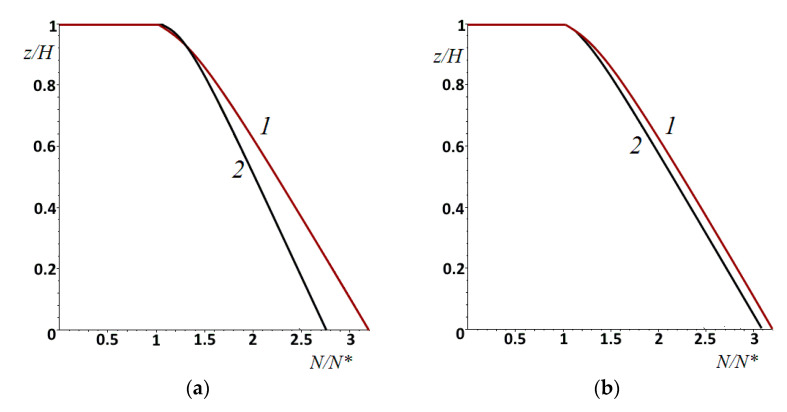
Displacement of the layer surface (wear kinetics). Effect of sliding velocity (**a**): $V^{\prime }=0.0208$ (curve 1) or $V^{\prime }=0.0625$ (curve 2), μ=0, c=5. Effect of friction (**b**): μ=0 (curve 1) or μ=0.3 (curve 2),$V^{\prime }=0.0625$, c=5. Effect of parameter c (**c**): *c* = 2, 5, or 10 (curves 1–3, respectively), μ=0 (curve 2), $V^{\prime }=0.0208$. ν=0.4, ${p_{n}}^{\prime }=4.2\cdot 10^{-2}$, $h^{\prime }=0.667$.

**Figure 8 materials-14-06513-f008:**
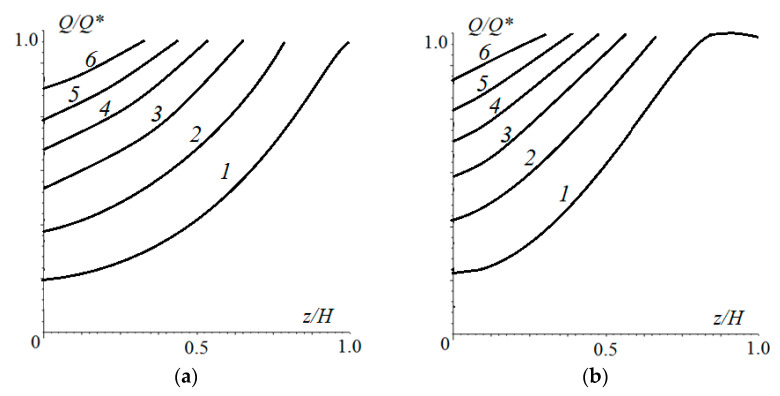
Distribution of damage within the elastic layer at different moments of time. Stochastic time load distribution varied from the average nominal pressure by ±20% (**a**) and ±80% (**b**) of average amplitude. N/N*=1, 1.1, 1.2, 1.3, 1.4, or 1.5 ((**a**), curves 1-6, respectively), N/N*=1, 1.07, 1.14, 1.21, 1.28, or 1.35 ((**b**), curves 1–6, respectively), ν=0.4, ${p_{n}}^{\prime }=4.2\cdot 10^{-2}$, $h^{\prime }=0.667$, $V^{\prime }=0.0208$.

**Figure 9 materials-14-06513-f009:**
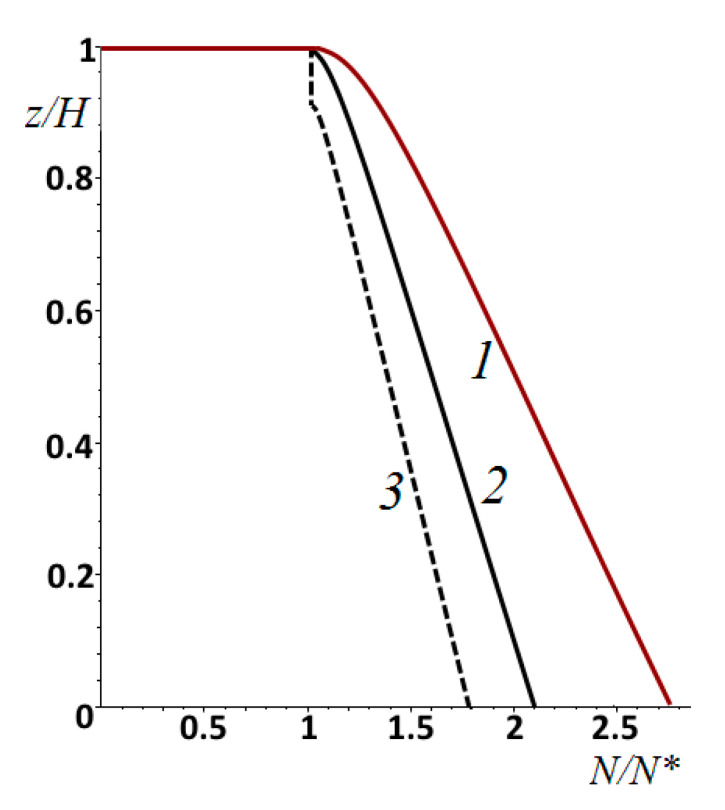
Displacement of the layer surface due to fatigue wear at different loading conditions: amplitude of nominal pressure was constant (curve 1) or varied by ±20% (curve 2) or ±80% (curve 3). ν=0.4, ${p_{n}}^{\prime }=4.2\cdot 10^{-2}$, $h^{\prime }=0.667$, $V^{\prime }=0.0208$.

## Data Availability

Data sharing is not applicable to this article.

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
