# Peer review of "Modeling of Fatigue Wear of Viscoelastic Coatings"

_materials, 2021, doi:10.3390/ma14216513_

Round 1

Reviewer 1 Report

In the paper “Modeling of Fatigue Wear of Viscoelastic Coatings,” the authors studied and modeling the fatigue wear of relatively compliant coatings made of viscoelastic materials (rubbers?) based on experiments made by other authors like Kragelsky and  Nepomnyashchy  [5] Mars, and Fatemi [10], Zhang et. al  [16]. Based on references [5] and [15] they used the fatigue fracture solution obtained by other authors [15].

The simulation approach of viscoelastic coating fracture, due to cyclic loading is very well made, taking into account the initial conditions as (i) calculation of contact pressure, (ii) calculation of internal stresses, and other algorithm aspects.

In actual condition, when the materials and experimental conditions and/or equipment(s) are not presented ( they were only referred to), consider that the paper is more suitable for journals like “Journal applied mathematics mechanics” or “Journal of mechanics and Physics of Solids” etc.

Specific comments are as follows:

  1. Because I cannot have access online and in English of the content of papers from references [1], [2], [3], [4], [5], [6], [17], [18], [19], [20] and [29], please  describe the material characteristics used in experiments (for instance viscoelastic properties), also the experimental equipment and conditions, on which it is based the  modeling and experimental study;
  2. Please verify the reference [9]. (see the link: https://pubs.acs.org/doi/abs/10.1021/bk-1985-0287.ch004 ).
  3. Please verify the volume, pages of reference [11]: Cadwell, S. M., Merrill, R. A., Sloman, C. M., & Yost, F. L. (1940). Dynamic fatigue life of the rubber. Rubber Chemistry and Technology13(2), 304-315.
  4. Line 54: I didn’t find the author “Saverin” at reference [15]. Please check and correct.
  5. Please complete/provide us with bibliographic references, if possible, from the last 5 years, that we can have access online.
  6. Is the thickness notation in Figure 2 correct? I think that is “ h” instead of H. Please confirm/ correct that.

In these conditions, I suggest that the paper could be published after you solved the above-mentioned comments/requirements.

Author Response

The authors are grateful for carefully reading the paper and valuable comments.

Below are the comments and our answers

Because I cannot have access online and in English of the content of papers from references [1], [2], [3], [4], [5], [6], [17], [18], [19], [20] and [29], please  describe the material characteristics used in experiments (for instance viscoelastic properties), also the experimental equipment and conditions, on which it is based the  modeling and experimental study;

The most important points concerning contact fatigue tests are collected in Appendix 1. Monograph [19] has been replaced by [34], which is in English.

Please verify the reference [9]. (see the link: https://pubs.acs.org/doi/abs/10.1021/bk-1985-0287.ch004 ).

Thanks, it's done

Please verify the volume, pages of reference [11]: Cadwell, S. M., Merrill, R. A., Sloman, C. M., & Yost, F. L. (1940). Dynamic fatigue life of the rubber. Rubber Chemistry and Technology13(2), 304-315.

There are two papers by these authors with the same title in different journals.

Line 54: I didn’t find the author “Saverin” at reference [15]. Please check and correct.

Thanks, it's done (Ref. [32]).

Please complete/provide us with bibliographic references, if possible, from the last 5 years, that we can have access online.

We include more references to Introduction

Is the thickness notation in Figure 2 correct? I think that is “ h” instead of H. Please confirm/ correct that.

Thank you very much! H is correct; h is changed to H in several figures.

Reviewer 2 Report

The submitted manuscript discusses the modeling of wear of viscoelastic coatings in the conditions of an intender being a friction pair with the coated surface. The authors presented a brief introduction, which needs some revision (see the detailed comments). In section 2, the authors presented subsequent steps of modeling assumed in this study. This section also needs improvements. In the next two sections, the theoretical fundamentals on viscoelasticity in the investigated friction pair and the formulation of the developed model were given. Sections 5 and 6 presents the results of calculations. In the latter section, the stochastic approach was used for a calculation of a fatigue wear. The proposed model have several deficiencies mentioned in the detailed comments. Moreover, the study is purely theoretical, without an experimental validation, which in the case of viscoelastic coatings and damage accumulation may lead to significant deviations in results. A deep revision of the manuscript is necessary before further consideration.

1) The authors should explicitly formulate their research problem at the end of the first section, indicating what exactly was done in the cited references and what novelty and improvements are obtained by the authors with respect to previously developed models. Moreover, it is likely to define a class of practical problems where the developed model may find an application.

2) Section 2 should be enriched by appropriate references and justifications, which will help a reader to understand the origin of the assumptions made by the authors.

3) The authors should describe what is “mutual effect”.

4) The formulated model does not consider any thermal components, which are expected to be significantly influencing on wear processes. If the authors consider a viscoelastic coating, there are at least two possible sources of heat: viscoelastic response of a coating and friction. Several studies consider this problem as the coupled thermomechanical one. If the authors do not consider thermal effects, the explicit assumptions with valid justification should be given in the manuscript.

5) Please comment on the assumed parameters of calculations in lines 221-222. On which basis they were selected and how they address to the properties of real coatings?

6) The literature is outdated in many cases and limited primarily to Russian authors. It is essential to improve the literature survey by recent developments of other teams outside Russia.

7) The conclusions are not supported by the obtained results. The authors should provide quantitative results in conclusions and exclude references from the last paragraph of the Conclusions section, since the provided conclusions are based on the cited references, but not on the obtained results in the study.

8) The experimental validation of the obtained computational results is necessary. The calculations can be performed based on the experimental results published in another study, the validation in the defined problem need to be provided.

9) The language of the manuscript need to be improved.

Author Response

The authors are grateful to the reviewer for his work and important comments.

Below are the comments and our answers

 The authors should explicitly formulate their research problem at the end of the first section, indicating what exactly was done in the cited references and what novelty and improvements are obtained by the authors with respect to previously developed models. Moreover, it is likely to define a class of practical problems where the developed model may find an application.

Thanks a lot for this comment.  End of section 1 is revised. In particular, the following text has been added:

This study is a logical continuation of modeling the fatigue wear of elastic layered bodies [36] and viscoelastic half-space [37].

Section 2 should be enriched by appropriate references and justifications, which will help a reader to understand the origin of the assumptions made by the authors.

Thanks, it's done

The authors should describe what is “mutual effect”.

We included the explanation:  “influence of neighboring indenters on contact pressures under a selected indenter” (Page 4, paragraph 3)

The formulated model does not consider any thermal components, which are expected to be significantly influencing on wear processes. If the authors consider a viscoelastic coating, there are at least two possible sources of heat: viscoelastic response of a coating and friction. Several studies consider this problem as the coupled thermomechanical one. If the authors do not consider thermal effects, the explicit assumptions with valid justification should be given in the manuscript.

Heat effects do not always occur during friction of rubbers (cases of rolling, or friction with an intermediate layer that levels adhesion, or slow sliding). Of course, frictional heating significantly changes the properties of rubbers; destruction of the material occurs. Under these conditions, the model proposed in this study is inapplicable.

In the Introduction, we added text about different wear mechanisms.

Please comment on the assumed parameters of calculations in lines 221-222. On which basis they were selected and how they address to the properties of real coatings?

We use dimensionless parameters that allow us to apply the analysis results to many cases of frictional loading of viscoelastic coatings. Explanations for the choice of Poisson's ratio and initial layer thickness have been added.

The literature is outdated in many cases and limited primarily to Russian authors. It is essential to improve the literature survey by recent developments of other teams outside Russia.

Thanks, it's done

The conclusions are not supported by the obtained results. The authors should provide quantitative results in conclusions and exclude references from the last paragraph of the Conclusions section, since the provided conclusions are based on the cited references, but not on the obtained results in the study.

The last paragraph, which contains links and is important from a physical point of view, has been moved from the conclusions to the main text of the paper. The conclusions are expanded.

The experimental validation of the obtained computational results is necessary. The calculations can be performed based on the experimental results published in another study, the validation in the defined problem need to be provided.

In this study, only the criterion of damage accumulation is taken from the experiments. The results obtained by the methods of solid mechanics are universal. The conclusions, as it is often in the case of modeling, are qualitative. There are many papers, including in well-known journals, where there is only modeling without experiments or direct references to them (for example https://doi.org/10.1016/j.wear.2021.203685  or  [36]). The presence of an incubation period has been proven in many studies, as described in the Introduction.

Reviewer 3 Report

This paper presents the results of modeling the fatigue wear of relatively compliant 64 coatings made of viscoelastic materials. Before considering it for publication the authors are required to revise it carefully as mentioned:

  1. The abstract should be organized, including the novelties.
  2. The introduction should be improved including new relevant works.
  3. Fig 1 should be withdrawn. It can be cited that’s all.
  4. All the parameters should be defined clearly under each equation.
  5. The authors should describe clearly the software used to plot Figs 4-5.
  6. English should be revised carefully (grammatical, typos,…).
  7. Did the authors study the convergence study compared with experimental analysis?
  8. The authors mentioned this model is validated with experimental but the validation does not exist.
  9. The resolution of the last figs should be improved. The last Figs are original?
  10. Some relevant works related to damage identification based on new techniques should be added to improve the introduction such as:

https://doi.org/10.3390/met11101616

https://doi.org/10.1016/j.compstruct.2021.114287

https://doi.org/10.3390/ma14195885 

https://doi.org/10.1016/j.compstruct.2020.112497

  1. The conclusion should be improved including more details and the advantage of the presented technique.

Author Response

The authors are grateful for carefully reading the paper and valuable comments.

Below are the comments and our answers

The abstract should be organized, including the novelties.

We drew attention to the fact that the proposed model is new. We also added the following phrase about results: Two types of load, constant and stochastically varying, are used in modeling and analysis. It was found that in the process of fatigue wear of the coating, its rate increases, then becomes constant.

The introduction should be improved including new relevant works.

Thanks, it's done

Fig 1 should be withdrawn. It can be cited that’s all.

Another reviewer found it necessary to write more about these studies. We did it in the Appendix. Figure 1 has been moved there.

All the parameters should be defined clearly under each equation.

All parameters and variables are described where they are entered in the text. We did not find any parameters not described. If we have not done this somewhere, we will be grateful for specifying the parameter and page. This is a very important point.

The authors should describe clearly the software used to plot Figs 4-5.

We have added a corresponding sentence at the end of Section 4

Did the authors study the convergence study compared with experimental analysis?

In this study, only the criterion of damage accumulation is taken from the experiments. The results obtained by the methods of solid mechanics are universal. The presence of an incubation period has been proven in many studies, as described in the Introduction.

The authors mentioned this model is validated with experimental but the validation does not exist.

As mentioned above, the criterion of damage accumulation used in the model can be considered experimentally proven. If the text contains an ambiguous phrase about this, please write in which paragraph. We will change it.

The resolution of the last figs should be improved. The last Figs are original?

The resolution is good enough. This is the effect of reducing the size of the picture when inserted into the text. We will provide originals for publication.

Some relevant works related to damage identification based on new techniques should be added to improve the introduction such as:

https://doi.org/10.3390/met11101616

https://doi.org/10.1016/j.compstruct.2021.114287

https://doi.org/10.3390/ma14195885

https://doi.org/10.1016/j.compstruct.2020.112497

Many thanks for the papers! We have included some of them.

The conclusion should be improved including more details and the advantage of the presented technique.

The Conclusions section has been supplemented with appropriate text.

Round 2

Reviewer 1 Report

I agree with the modifications and completions made by the authors, and as a result, I agree with the publication of the paper.

Author Response

The authors are grateful for the review of their paper.

Reviewer 2 Report

The authors performed most of the necessary corrections and provided a wide explanation to the comments from the first round of review. The only comment which was left without changes in the manuscript is comment 2 from the previous review report:

Section 2 should be enriched by appropriate references and justifications, which will help a reader to understand the origin of the assumptions made by the authors.

I advise the authors to perform appropriate explanations in the manuscript.

Author Response

The authors are grateful to the reviewer for his attention to their research. The following text has been added to the beginning of Section 2:

Under cyclic loading of the surface, which occurs with relative displacements of rough bodies, an inhomogeneous cyclic field of internal stresses with high amplitude values arises in the surface layer, which is the reason for the accumulation of damage in this layer. A macroscopic approach for studying the damage accumulation process was originally developed in relation to standard fatigue tests, and is described, for example, in [38]. It was found by Goryacheva [34] that this approach can be used for modeling fatigue wear. 

Ref. [38] is new.

Reviewer 3 Report

Accept in present form. No Further comments. 

Author Response

(The authors gave the same response as above.)
